# MedJourney: Counterfactual Medical Image Generation by Instruction-Learning from Multimodal Patient Journeys

## Abstract

Rapid progress has been made in instruction-learning for image editing with natural-language instruction, as exemplified by InstructPix2Pix. In biomedicine, such counterfactual generation methods can help differentiate causal structure from spurious correlation and facilitate robust image interpretation for disease progression modeling. However, generic image-editing models are ill-suited for the biomedical domain, and counterfactual medical image generation is largely underexplored. In this paper, we present *MedJourney*, a novel method for counterfactual medical image generation by instruction-learning from multimodal patient journeys. Given a patient with two medical images taken at different time points, we use GPT-4 to process the corresponding imaging reports and generate a natural language description of disease progression. The resulting triples (prior image, progression description, new image) are then used to train a latent diffusion model for counterfactual medical image generation. Given the relative scarcity of image time series data, we introduce a two-stage curriculum that first pretrains the denoising network using the much more abundant single image-report pairs (with dummy prior image), and then continues training using the counterfactual triples. Experiments using the standard MIMIC-CXR dataset demonstrate the promise of our method. In a comprehensive battery of tests on counterfactual medical image generation, MedJourney substantially outperforms prior state-of-the-art methods in instruction image editing and medical image generation such as InstructPix2Pix and RoentGen. To facilitate future study in counterfactual medical generation, we plan to release our instruction-learning code and pretrained models.

## 1 Introduction

Biomedical data is inherently multimodal, comprising physical measurements and natural-language narratives. In particular, biomedical imaging represents an important modality for disease diagnosis and progression monitoring. Counterfactual medical image generation seeks to answer the "what if" question in biomedical imaging Cohen et al. (2021); Sanchez & Tsaftaris (2022). E.g., given a radiology image of a cancer patient, what would the image look like if the cancer has undergone specific progression? Such capabilities can potentially make image interpretation more explainable and robust, by revealing the underlying causal structure as well as spurious correlation.

Existing methods for counterfactual medical image generation, however, are generally limited to modeling simple image class change. I.e., how would an image change to be classified as a different category Cohen et al. (2021); Sanchez & Tsaftaris (2022), as studied extensively in adversarial learning Zhang et al. (2019); Madani et al. (2018b). Such restricted counterfactual image generation can be viewed as image editing with a fixed set of predefined class changes. Recently, there has been rapid progress in image editing with arbitrary natural-language instruction, as exemplified by InstructPix2Pix (Brooks et al., 2023). However, these models are trained using generic images and text, which makes them ill-suited for the biomedical domain. There have been recent attempts to adapt image generation to the radiology domain, as exemplified by RoentGen (Chambon et al., 2022a). But they condition image generation on text description only, rather than on the prior image and counterfactual conditions, thus are not well suited for counterfactual image generation. In general, unconstrained counterfactual medical image generation remains largely unexplored.

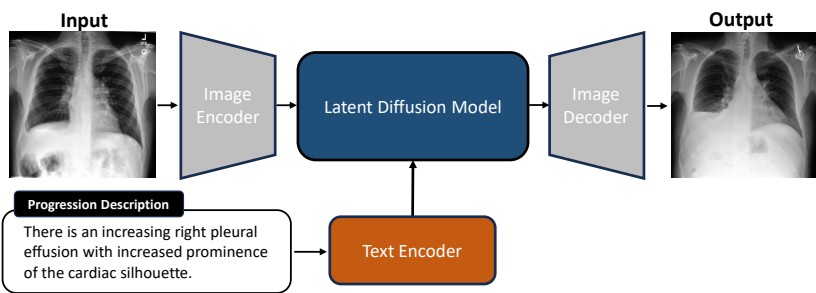

Figure 1: The general architecture of *MedJourney* for counterfactual medical image generation. Given the prior image, and a text description of patient disease progression, *MedJourney* would generate the counterfactual image that *maximally reflects the prescribed changes while minimizing deviation from the original image.*

In this paper, we present *MedJourney*, a novel method for counterfactual medical image generation by instruction-learning from multimodal patient journeys. Given two medical images taken at different time points for a given patient, detailed information about the disease progression is readily available in their corresponding reports, but manually synthesizing such "image-editing instruction" from the reports is expensive and time-consuming. To scale instruction-following data generation, we use GPT-4 to generate a natural language description of disease progression from the two corresponding reports. We then apply the resulting triples (prior image, progression description, new image) to train a latent diffusion model for counterfactual medical image generation.

Compared to single image-report pairs, image time series are relatively scarce. *e.g.*, there are over three hundred thousand radiology image-text pairs in MIMIC-CXR Johnson et al. (2019), the largest publicly available de-identified medical multimodal dataset, but only about ten thousand counterfactual triples can be generated from the image-report time-series data. Therefore, instead of directly learning the counterfactual triples, we introduce a curriculum learning scheme that first pretrains the diffusion model using the much more abundant single image-report pairs using a dummy prior image, and then continues training using the counterfactual triples. We conduct extensive experiments on MIMIC-CXR. On a battery of tests for counterfactual medical image generation, MedJourney substantially outperforms prior state-of-the-art methods in instruction image editing and medical image generation such as InstructPix2Pix Brooks et al. (2023) and RoentGen Chambon et al. (2022a). We summarize our main contributions below:

- We propose *MedJourney*, which is the first method for counterfactual medical image generation that can closely follow arbitrary natural-language descriptions of disease progression to generate counterfactual images of high quality.

- We introduce a novel way for adapting general-domain text-to-image generation to counterfactual medical image generation by leveraging GPT-4 to produce the first instruction-following dataset at scale from multimodal patient journeys.

- We explore an extensive suite of tests for evaluating counterfactual medical image generation, such as pathology, race, age, and spatial alignment.

- We conduct extensive experiments on MIMIC-CXR and show that *MedJourney* substantially outperforms state-of-the-art methods such as Instruct-Pix2Pix and RoentGen on counterfactual medical image generation.

- We plan to release our instruction-learning code and pretrained models to facilitate future study in counterfactual medical image generation.

## 2 RELATED WORKS

**General Image Generation and Editing.**    Some pioneering works like Generative Adversarial Networks (GAN) (Goodfellow et al., 2014) and Variational Auto-Encoder (Kingma et al., 2019) had precipitated a surge of image generation works in general domain (Mao et al., 2017; Karras et al., 2019; Yoon et al., 2019; Higgins et al., 2016; Yang et al., 2017; Karras et al., 2019). Most recently, diffusion-based model arises and demonstrates promising image generation performance (Ho et al., 2020; Nichol & Dhariwal, 2021). Later on, a latent diffusion model (LDM) is consequently proposed which significantly unleashes the power to generate images of high quality and resolution (Rombach

et al., 2022). Built on top of LDM are a number of works to make it more spatial-aware (Yang et al., 2022; Li et al., 2023), controllable (Zhang et al., 2023a; Mou et al., 2023; Huang et al., 2023) and customizable (Ruiz et al., 2022). Going beyond text-to-image generation, text-based image editing recently drew increasing attention. A number of works propose to alter a small portion of visual contents in a given image, which can be the image style (Meng et al., 2022; Zhang et al., 2023c), local objects(Meng et al., 2022; Hertz et al., 2022; Kawar et al., 2023), *etc*. To build a more natural image editing interface, Instruct-Pix2Pix (Brooks et al., 2023) replaces the plain texts (*e.g.*, "a dog") with instructions (*e.g.*, "change cat to dog"). The authors developed an effective way to compose synthetic paired images using GPT-4 (**?**) and Prompt2Prompt (Hertz et al., 2022). Our work is inspired by Instruct-Pix2Pix but goes further to develop an instructed image generation model for medical images, particularly for chest X-ray images. Rather than synthesizing training data, we curate a good number of real paired data and develop a reliable model that not only cares about image quality but also the genuine emulation of real patient journeys.

**Medical Image Generation.** While there has been a large amount of work for image generation in the general domain, the medical domain is under-explored. Some earlier works used GAN for synthesizing different types of medical images (Costa et al., 2017; Madani et al., 2018b;a; Zhang et al., 2018; Zhao et al., 2018; Yi et al., 2019). To address the problem of limited training data, the authors in Madani et al. (2018a) and Madani et al. (2018b) proposed to use GAN to generate X-ray images for data augmentation. In Zhang et al. (2019), the generation process is decomposed into sketch and appearance generation which facilitates the generation of diverse types of medical images, such as X-Ray, CT and MRI, *etc*. With the rise of latent diffusion models (LDMs), the quality of generated medical images is significantly improved (Chambon et al., 2022b;a; Packhäuser et al., 2023; Schön et al., 2023). Beyond 2D images, it is further applied for 3D image synthesis (Dorjsembe et al., 2023; Khader et al., 2023). Most recently, some works explored a few ways of unifying medical reports and image generation into a single framework by leveraging a sequence-to-sequence model (Lee et al., 2023a) or a pre-trained large language model (LLM) (Lee et al., 2023b). All these works studied the potential of utilizing LDMs for medical text-to-image generation and mitigating the scarcity of real medical data. Our work significantly differs from existing studies in its goal to emulate the medical journey of individual patients by leveraging temporally paired multi-modal data (*i.e.* medical reports and images). We advocate for the development of a model capable of "editing" a source image in strict adherence to provided instructions that chronicle changes over a specific time frame. To the best knowledge, we are the first to study the instructed image generation in the medical domain, and strongly envision its prospects for counterfactual analysis.

**Counterfactual Analysis in Medicine.** Counterfactual analysis serves as a crucial way to make image interpretation more understandable and robust. Typically, for a medical image classifier, different visualization tools (Simonyan et al., 2013; Zhou et al., 2016; Selvaraju et al., 2017) can be employed to interpret the model and spot the spurious correlations (Kim et al., 2019). Later on, a number of works studied the way of generating counterfactual images to probe the existing image classifiers (Thiagarajan et al., 2022; Lenis et al., 2020; Fontanella et al., 2023; Major et al., 2020; Cohen et al., 2021; Atad et al., 2022; Bedel & Çukur, 2023). However, these approaches largely rely on gradient-based methods that are limited by their dependence on pre-determined classification labels as targets. In contrast, our approach also supports image modification but is designed to take arbitrary textual instructions as inputs, favoring much more flexible and customizable medical image editing. This makes our model a powerful image-editing tool for counterfactual analysis and, more importantly, helps to emulate the temporal disease transition of a patient.

## 3 DATASET AND PREPROCESSING

We use the standard MIMIC-CXR dataset Johnson et al. (2019) in our study, which contains 377,110 image-report pairs from 227,827 radiology studies. A patient may have multiple studies, whereas each study may contain multiple chest x-ray (CXR) images taken at different views. In this work, we only use posteroanterior (PA), the standard frontal chest view, and discard AP and lateral views. This results in 157,148 image-text pairs. We follow the standard partition and use the first nine subsets (P10-P18) for training and validation, while reserving the last (P19) for testing. We then identify patient journey and generate 9,354, 1,056, 1,214 counterfactual triples for train, validation, test, respectively, following the procedure below.

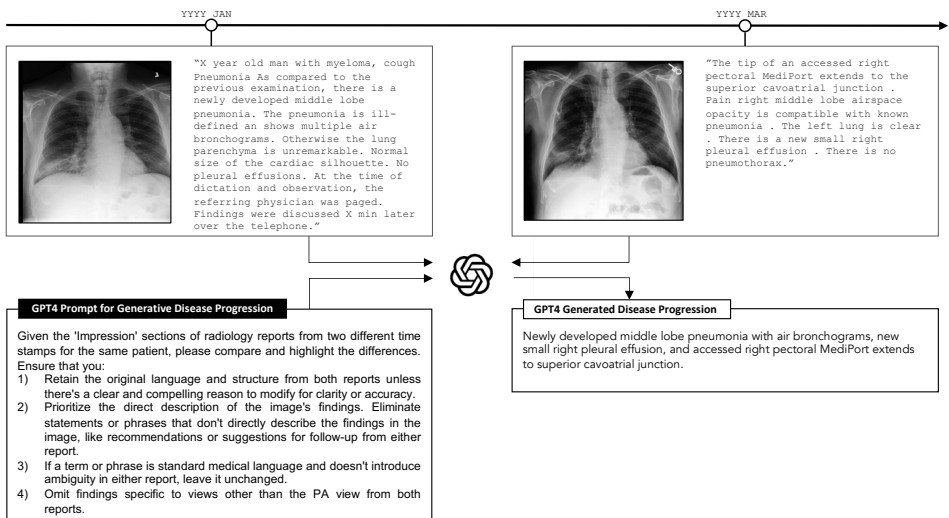

Figure 2: *MedJourney* uses GPT-4 to generate instruction-following data from multimodal patient journeys. Top: two images taken at different time points for a patient and their corresponding reports. Bottom left: GPT-4 prompt for synthesizing disease progression from the two reports. Bottom right: Example disease progression generated by GPT-4.

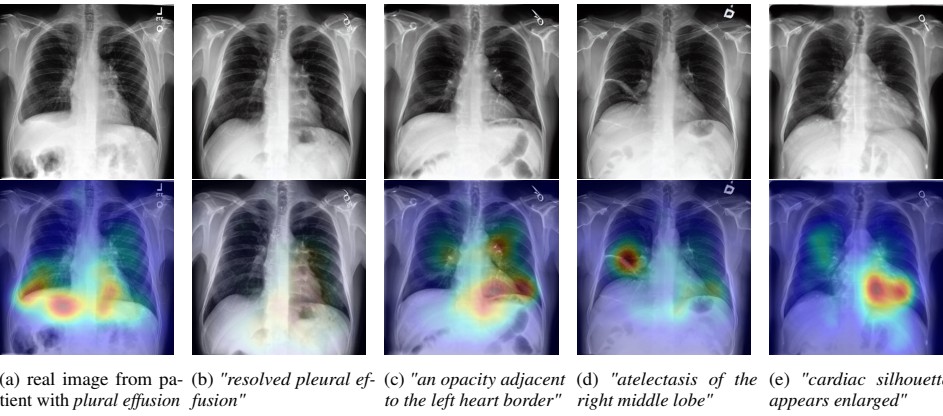

(a) real image from pa-
tient with *plural effusion*

(b) *"resolved pleural ef-
fusion"*

(c) *"an opacity adjacent
to the left heart border"*

(d) *"atelectasis of the
right middle lobe"*

(e) *"cardiac silhouette
appears enlarged"*

Figure 3: Example counterfactual generation by *MedJourney*. Top: source image and counterfactual images given various progression description. Bottom: attention map for the italicized description by a state-of-the-art pathology classifier Cohen et al. (2021). *MedJourney* generates counterfactual images that generally reflect the prescribed progression well.

**Identify patient journey.** We order studies with PA view for the same patient and select consecutive pairs as candidates for generating counterfactual triples.

**Image registration.** To mitigate unwanted artifacts stemming from varying image positions and angles across studies, we perform standard registration, using the SimpleITK toolkit Beare et al. (2018) (see Appendix).

**Data filtering.** We prioritize studies with at least one finding as identified by CheXpert and filter the rest. There is a nontrivial portion of MIMIC-CXR images with mislabeled views, which can lead to severe hallucinations. We further filter out image pairs for which the registration score is below a threshold, thus indicating misaligned views.

**Text preprocessing.** Radiology reports typically contain a "Findings" section with detailed obser-vations and an "Impression" section with a concise summary of the most salient findings. In this paper, we focus on the Impression section and explore using GPT-4 to clean the text (Figure 2) before generating instruction-following data.

# 4 METHOD

## 4.1 PROBLEM DEFINITION

Given the prior image $\mathbf{I}_P \in \mathcal{R}^{H \times W \times 3}$ and a progression description $\mathbf{D}$, the goal of counterfactual medical image generation is to learn a generative model $Gen$ for generating the counterfactual image $\mathbf{I}_C \in \mathcal{R}^{H \times W \times 3}$, *i.e.*, $\mathbf{I}_C = Gen(\mathbf{I}_P, \mathbf{D})$.

Compared to text-only medical image generation models such as RoentGen (Chambon et al., 2022a), our problem setting is very different: the generation description $\mathbf{D}$ describes the counterfactual changes from the prior image to the reference image, rather than the description of the reference image itself. This means that the model inputs both textual description and the prior image and is expected to produce a new image that reflects the prescribed changes while preserving other invariant aspects.

## 4.2 MEDJOURNEY

We build upon the state-of-the-art latent diffusion model (LDM) (Rombach et al., 2022), which comprises two components:

- An image autoencoder based on VQGAN that can project an image to $z$ in latent space and then reconstruct the image from $z$.

- A UNet Ronneberger et al. (2015) comprised of transformer and convolutional layers for performing latent diffusion in the latent space $z$, starting from a random Gaussian noise.

By default, LDMs only take text prompts as the input for image generation. In *MedJourney*, we extend LDM to condition generation on both text (progression description) and an image (prior image). Given training counterfactual triples ($I_P$, $D$, $I_C$), *MedJourney* adopts the standard LDM loss while concatenating the prior image encoding with the latent diffusion state:

$$\min_\theta \mathcal{L} = \mathbb{E}_{z_t, \epsilon \sim \mathcal{N}(0,1), t} \left[ ||\epsilon - f_\theta(z_t, t, \mathbf{E}(D), \mathbf{E}(I_P))||_2^2 \right] \tag{1}$$

where $t$ represents the diffusion time point, $\mathbf{E}(D)$ is the description embedding and $\mathbf{E}(I_P)$ is the prior image embedding. Following Instruct-Pix2Pix (Brooks et al., 2023), we concatenate the image condition $\mathbf{E}(I_P)$ with $z_t$ and feed them into the LDMs, while using attention layers to cross-attend the text condition $\mathbf{E}(D)$. The objective is to learn a denoising network $f_\theta$ that can reconstruct the noise $\epsilon$ given the noisy variant $z_t$ of latent representation for the reference image at timestep $t \in \{1, ..., T\}$. Once trained, the model can input a prior image and textual description to generate the reference image.

InstructPix2Pix (Brooks et al., 2023) uses CLIP (Radford et al., 2021) for the image and text encoders, which may be suboptimal for the biomedical domain. We explore replacing CLIP with Biomed-CLIP (Zhang et al., 2023b), which was pretrained on image-text pairs extracted from biomedical papers. BiomedCLIP also accepts much larger context length (increased from 77 to 256) that is more suited for clinical reports (vs general domain image captions). We introduce a learnable linear projection layer to bridge the new text encoder with UNet.

Initially, we trained *MedJourney* using only the counterfactual triples. Given their relative scarcity, the model easily overfits and hallucinations abound.

Consequently, we propose two-stage curriculum learning to leverage the much more abundant single image-text pairs:

- **Stage 1: Pretraining.** We first train $f_\theta$ using all image-text pairs in MIMIC-CXR training with the prior image set to a dummy image $I_P = \{128\}^{H \times W \times 3}$ of constant value 128.

- **Stage 2: Instruction Tuning.** We then fine-tune *MedJourney* with the counterfactual triples with a real prior image $I_P$.

## 4.3 EVALUATION METRICS

For evaluation, we assemble a set of instances each comprising two images taken at different times for a patient, their corresponding reports, and the GPT-4 generated progression description

by synthesizing the two reports. At test time, *MedJourney* takes the first image and progression description and outputs the counterfactual image. The second image is used as the reference standard.

Ideally, the counterfactual image should accurately reflect the prescribed changes in disease pathology while minimizing deviation from the prior image. We propose the Counterfactual Medical Image Generation (CMIG) score that balances accuracy and feature retention measurements. Given accuracy measurements $a_1, \cdots, a_n$ and feature retention measurements $f_1, \cdots, f_m$, we first compute their respective geometric means $\bar{a} = \sqrt[n]{\prod_i a_i}$ and $\bar{f} = \sqrt[m]{\prod_j f_j}$, and then return the CMIG score as their geometric mean $\sqrt{\bar{a} \cdot \bar{f}}$. This ensures that the final score is not skewed to either aspect, as they are both important for counterfactual generation. We choose geometric mean as it is robust to results of varying scales. The CMIG scoring scheme is general. In this paper, we adopt a simple instantiation using a pathology classifier for accuracy, a race classifier and an age classifier for feature retention.

- **Pathology Classifier:** We use the DenseNet-121 model from the XRV collection Cohen et al. (2021), a state-of-the-art image classifier for CheXpert findings Irvin et al. (2019). We run it on the counterfactual image to obtain the predicted pathology finding labels. Following RoentGen Chambon et al. (2022a) to enable head-to-head comparison, we run CheXpert Irvin et al. (2019) on the corresponding report of the reference image to obtain reference pathology labels, focusing on the five most prevalent findings (Atelectasis, Cardiomegaly, Edema, Pleural Effusion, Pneumothorax), and then compute AUROC of the predicted labels against the reference ones.

- **Race Classifier:** Similarly, we use the state-of-the-art image classifier for race Gichoya et al. (2022) to generate predicted labels on the system-generated counterfactual images, gold race information from MIMIC as reference labels, and compute AUROC.

- **Age Classifier:** We use the state-of-the-art DNN model Ieki (2022) to predict age from the image, the anchor age information from MIMIC as reference, and return Pearson correlation. MIMIC only contains patient data within a short duration, so the two images are taken when the patient is approximately at the anchor age.

## 5 EXPERIMENTS

### 5.1 EXPERIMENTAL SETUP

**Implemetantion details.** We use the pre-trained Stable Diffusion v1.5 model [1] and BiomedCLIP [2] for initialization. Similar to Instruct-Pix2Pix, we keep the text encoder frozen and add a learnable linear projection layer on top of BiomedCLIP text encoder. *MedJourney* is trained in two stages. In Stage 1 (pretraining), we use 69,846 single image-text pairs and train the model with $8 \times 40$GB A100 GPUs for 200 epochs in 36 hours. In Stage 2 (instruction tuning), we continue training the model for another 128 epochs using the counterfactual triples in another 23 hours. For both stages, the image resolution is set to $256 \times 256$, and horizontal flip and random crop are used for data augmentation. We use AdamW (Loshchilov & Hutter, 2019) and a fixed learning rate of $1e^{-4}$ with batch size 32.

**Baseline systems.** We compare *MedJourney* to state-of-the-art representative works. $(i)$ **Stable Diffusion** (SD) (Rombach et al., 2022): We use the target Impression as the text prompt. Notably, we need to prepend "a photo of chest x-ray" to the prompt to generate meaningful results. $(ii)$ **RoentGen** (Chambon et al., 2022a): state-of-the-art text-only medical image generation model. Similarly, target Impression is used as the text prompt, in line with RoentGen's training. $(iii)$ **InstructPix2Pix** (IP2P) (Brooks et al., 2023): state-of-the-art instruction-tuned image-editing model for the general domain.

### 5.2 MAIN RESULTS

Table 1 compares *MedJourney* with prior state-of-the-art systems. *MedJourney* substantially outperforms other systems across all aspects. Not surprisingly, general-domain SD and IP2P models perform extremely poorly across the board. E.g., SD's pathology accuracy is close random. As expected,

---

[1]`https://huggingface.co/runwayml/stable-diffusion-v1-5/resolve/main/`
`v1-5-pruned-emaonly.ckpt`
[2]`https://huggingface.co/microsoft/BiomedCLIP-PubMedBERT_256-vit_base_patch16_224`

Table 1: Comparison of test results for counterfactual medical image generation. *Pathology AUC* measures accuracy (how well the generated image reflects the relevant pathology findings). *Race AUC* and *Age Pearson Correlation* gauge feature retention (how well the generated image retains invariant features such as race and age - age rarely changes given that MIMIC only contains data within a short duration for each patient). Our proposed CMIG score returns the geometric mean of the two respective geometric means for accuracy and feature retention results. This ensures that the final score is not skewed to either aspect, as they are both important for counterfactual generation. We choose geometric mean as it is robust to results of varying scales.

| Model | Pathology AUC | Race AUC | Age Pearson Corr. | CMIG Score |
|---|---|---|---|---|
| SD (Rombach et al., 2022) | 49.90 | 77.13 | 2.73 | 18.14 |
| IP2P (Brooks et al., 2023) | 58.10 | 78.25 | 17.82 | 42.12 |
| RoentGen (Chambon et al., 2022a) | 79.61 | 84.71 | 28.91 | 66.08 |
| MedJourney (Ours) | 80.54 | 97.22 | 79.38 | **83.23** |

(a) Source Image     (b) SD     (c) IP2P     (d) RoentGen     (e) MedJourney

**Progression Description**

There is an increasing right pleural effusion with increased prominence of the cardiac silhouette. No evidence of left pleural effusion. Increased prominence of central pulmonary vessels is observed, with no evidence of peripheral venous congestion.

Figure 4: Example of precision control of changes as exhibited by *MedJourney*. From left to right, an example prior image and counterfactual generated by various models with the progression description below.

RoentGen performs much better as it fine-tunes SD on MIMIC-CXR data. However, RoentGen only learns generic text-to-image generation that is not conditioned on the prior image, thus it is incapable of preserving patient-specific invariants such as race and age. By contrast, *MedJourney* excels in both pathology accuracy and feature retention, surpassing RoentGen in the CMIG score by over 17 points.

Figure 3 shows example counterfactual generations by *MedJourney* and the corresponding attention maps highlighted by a state-of-the-art pathology classifier Cohen et al. (2021). *MedJourney* demonstrates precision controls that generally reflect the prescribed progression well. Figure 4 contrasts *MedJourney* generation with others. There are two prominent progressions in the description: increased cardiomegaly and right pleural effusion. Not surprising, general-domain SD can't produce realistic CXR images. InstructPix2Pix generally preserves patient-specific features but can't execute the prescribed changes. Moreover, the overall quality of the generated image is much lower compared to the original image. The RoentGen generation is better than SD and InstructPix2Pix, but it still misses key pathology changes. The pleural effusion is supposed to reside at the right side of the lung (left from viewer's perspective), but the generated image shows pleural effusion on both sides. Moreover, the gender of the patient is altered and a device is added to the chest. By contrast, *MedJourney* demonstrates good instruction-following capability by presenting increased cardiomegaly and right pleural effusion while preserving patient invariants such as body build, gender.

## 5.3 ABLATION STUDY

Table 2 shows comprehensive ablation results for *MedJourney*.

**Image registration.** There is clear benefit in performing registration to align the image pair during training, as can be seen by comparing the top and bottom four rows in Table 2. E.g., the race AUC is improved by over eight points after registration for the two-stage model. Similarly for age. Image registration alleviates spatial misalignment, thus reducing the risk for the model to overfit to spurious correlation.

**Impression *v.s.* GPT-4 generation.** As we showed in Fig. 2, the Impression section can be noisy and fragmented. Moreover, it might miss some details of disease progression. GPT-4 generated description is usually more succinct (averaging 146 characters v.s. 179) and more naturally phrased

Table 2: Ablation study on various model settings: using target Impression vs GPT-4 synthesized description; using registration or not; using two-stage curriculum learning or not.

| Description | Registration | Two-Stage | Pathology AUC | Race AUC | Age Pearson Corr. | CMIG Score |
|---|---|---|---|---|---|---|
| Impression | No | No | 79.83 | 94.23 | 75.39 | 79.99 |
| GPT-4 | No | No | 80.71 | 94.76 | 76.78 | 80.76 |
| Impression | No | Yes | 82.80 | 87.81 | 47.13 | 75.43 |
| GPT-4 | No | Yes | 81.35 | 88.58 | 55.12 | 76.38 |
| Impression | Yes | No | 78.06 | 98.67 | 82.73 | 82.70 |
| GPT-4 | Yes | No | 78.22 | 98.70 | 82.96 | 82.83 |
| Impression | Yes | Yes | 80.46 | 96.37 | 79.16 | 82.96 |
| GPT-4 | Yes | Yes | 80.54 | 97.22 | 79.38 | 83.23 |

Table 3: Comparison of test AUROC for a state-of-the-art pathology classifier on counterfactual images for the five most prevalent conditions (Atelectasis, Cardiomegaly, Edema, Pleural Effusion, Pneumothorax), as well as the KL Divergence for label distribution (lower the better).

| Source | At. | Ca. | Ed. | Ef. | Px. | Mean | KL Divergence |
|---|---|---|---|---|---|---|---|
| SD (Rombach et al., 2022) | 49.65 | 50.79 | 54.48 | 45.31 | 49.28 | 49.90 | 92.65 |
| IP2P (Brooks et al., 2023) | 57.38 | 53.70 | 62.59 | 58.85 | 57.97 | 58.10 | 60.15 |
| Roentgen (Chambon et al., 2022a) | 70.05 | 77.85 | 78.66 | 86.20 | 85.26 | 79.61 | 40.74 |
| MedJourney (Ours) | 72.08 | 73.76 | 88.64 | 88.11 | 80.12 | 80.54 | 10.9 |

and understandable. As can be seen In Table 2, using GPT-4 generated description consistently improves model performance.

**One-stage *v.s.* two-stage training.** Interestingly, without registration, two-stage training actually produces worse results. Moreover, two-stage training leads to better pathology accuracy, at the expense of feature retention. We hypothesize that the two-stage model might learn to rely more on text description after the additional stage of training using single image-text pairs. Using registration in the second stage can substantially mitigate such degradation ($\sim$6 pts drop *v.s.* $\sim$2 pts drop on race AUC), potentially because the calibration eases the model to learn the transition of the reference image from the prior image. In the end, Overall, the best performance is attained using both registration and two-stage training.

## 5.4 MODEL INSPECTION

### 5.4.1 PATHOLOGY ACCURACY

In Table 3, we report more fine-grained results over the five prevalent pathology findings. Interestingly, we find that RoentGen has overall worse performance and fluctuates substantially at individual category. We hypothesize that the pathology classifier may not perform equally well for all categories. Since we use the pathology classifier to generate the predicted labels from the counterfactual image, but use CheXpert to generate the reference labels from the target report, the varying performance of the pathology classifier might create confounding results. We thus investigate another way to asssess pathology accuracy by applying the pathology classifier to both the counterfactual and reference images, and computing the KL-divergence (lower the better) over the predicted scores for individual categories (see Appendix). As can be seen in Table 3, *MedJourney* counterfactual images actually have much lower KL-divergence compared to the reference images, while applying the same pathology classifier. This indicates that the standard pathology accuracy evaluation in RoentGen might have inadvertently inflate its performance.

### 5.4.2 ANATOMY FEATURE RETENTION

In addition to feature retention of patient's age and race, we further investigate how well our model preserves the anatomy layout of the prior image. We use a segmentation model (Lian et al., 2021) to identify the major components in the image and compare the counterfactual segmentation with the reference one. As shown in Fig. 5, the *MedJourney* counterfactual is generally well aligned with the reference, whereas other models often perform much worse. In Table 4, we further obtain quantitative results by computing the Dice score (Sudre et al., 2017) averaged over the test set. *MedJourney* demonstrates clear superiority compared to all other methods.

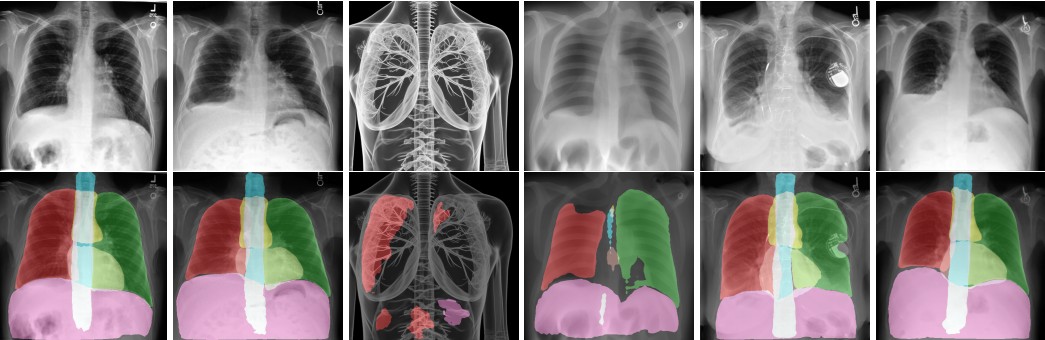

Figure 5: Comparison of the reference and counterfactual images in segmentation output by a state-of-the-art segmentation model. Top: image; bottom: segmentation output. Columns (from left to right): example prior image, reference image, images generated by Stable Diffusion, InstructPix2Pix, RoentGen and MedJourney (ours), respectively. We show masks for six different components including left lung, right lung, heart, facies diaphragmatica, mediastinum and spine.

| Model | Dice |
|---|---|
| SD | 1.37 |
| IP2P | 38.33 |
| RoentGen | 67.38 |
| Reference Image | 74.04 |
| MedJourney | 81.05 |

(a) Duplicated Organs  (b) Duplicated Ribs

Table 4: Comparison of segmentation concordance between reference and counterfactual.

Figure 6: Hallucinations of (a) organs and (b) ribs are observed in the earlier version (left) and fixed in the final version of our *MedJourney* (right). See Sec. 5.4.3 for the solutions.

### 5.4.3 HALLUCINATIONS

In the early development of *MedJourney*, we observed severe hallucinations, notably duplicated organs and ribs (left part of Fig. 6 (a) and (b)). The root causes included mismatched viewpoints of training pairs and resolution discrepancies between training and evaluation datasets. Specifically, (a) The duplicated-organs problem stemmed from mixing front and side views in training, which was resolved after data cleaning. (b) The duplicated-ribs problem stemmed from the disparity in image resolution between training and evaluation ($256 \times 256$ for training *v.s.* $512 \times 512$ for evaluation), which was resolved after we used $256 \times 256$ for both. Unlike the general image domain, we note that subtle differences in settings may lead to problematic degradation in medical image generation.

## 6  DISCUSSION

The initial results with *MedJourney* are promising, but much remains to be explored. Upon close inspection, we have identified image resolution as a potential cause for certain recurring errors (e.g., failure to generate very fine-grained changes). We are yet to explore more powerful image and text encoders for initialization, as well as full fine-tuning. MIMIC-CXR only features emergency medicine, which limits the learning of *MedJourney*. Finally, there are other accuracy and feature retention measurements that can be potentially incorporated for more comprehensive evaluation.

## 7  CONCLUSION

We propose *MedJourney*, the first general approach for counterfactual medical image generation with arbitrary natural-language instruction. *MedJourney* takes inspiration from general-domain image-editing methods such as InstructPix2Pix, and generates instruction-following data from multimodal patient journeys using GPT-4. We conduct extensive experiments on MIMIC-CXR and show that *MedJourney* substantially outperforms existing state-of-the-art methods on a battery of tests for counterfactual medical image generation. Future directions include: training *MedJourney* with higher image resolution; improving *MedJourney* curriculum learning; and exploring other medical domains and image modalities.

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

# A  APPENDIX

## A.1  DATASET DETAILS

The MIMIC-CXR dataset used in our work consists of 377,110 image-report pairs from 227,827 radiology studies conducted at Beth Israel Deaconess Medical Center. The dataset is partitioned into ten subsets (P10-P19). For training and validation, we employ the first nine subsets (P10-P18), while reserving the last (P19) for testing. To the end, our training set comprises 9,354 image-report pairs from P10-P18 for training, with an additional 1,056 pairs allocated for validation. The test set includes 1,214 pairs from P19. We elaborate the step-by-step data preprocessing below. Furthermore, we employ 69,846 posterior-anterior (PA) view images from P10-P18 for the initial stage of training.

In Fig. 7, we calculate the age and race statistics for our train/dev/test sets. It is clearly shown that some data bias happens in both age and race – the patients are skewed to elder and white people.

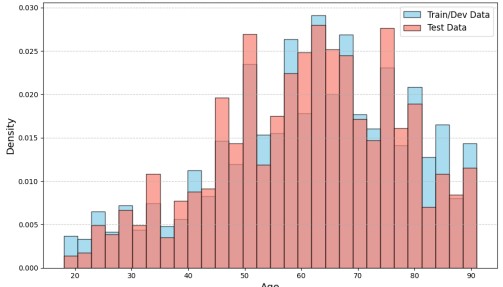 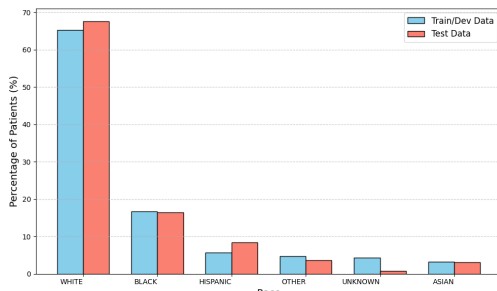

(a) Age distribution in train and test data sets. Note the older age distribution dominance.

(b) Race distribution in train and test data sets. White is the dominant class.

Figure 7: Distribution of age and race in our data sets, providing insight into the demographic composition.

## A.2  MATHEMATICAL FOUNDATION

We explain below the mathematical foundation for the metrics used in our main submission.

For an image $I$, the classifier $C$ yields a predicted label vector $C(I)$, contrasted against the CheXpert's textual-based label $L(I)$. The soft label AUC is defined as $\text{AUC}_{\text{soft label}} = C(I) \times L(I)$.

To objectively differentiate between real and synthesized images, we employ the KL divergence between $p = C(I_{\text{real}})$ and $q = C(I_{\text{synthesized}})$:

$$\text{KL divergence} = \sum_i p(i) \log \frac{p(i)}{q(i)}$$

## A.3  ADDITIONAL ABLATION STUDIES

**Role of text encoder.**  We further study the impact of the text encoder used to encode the instructions. By default, our model uses BiomedCLIP as the text encoder considering it is pre-trained with multi-modal medical data. Here, we replace it with CLIP and PubMedBERT (Gu et al., 2021) text encoder, respectively. For fair comparison, we use one-stage training for all models. In Table 5, it is shown that BiomedCLIP outperforms CLIP and PubMedBERT on average, which demonstrates a better trade-off between pathology and feature preservation. In particular, since BiomedCLIP has a better understanding of the textual instructions in medical domains, it achieves substantially better performance in pathology. Compared with CLIP, PubMedBERT is pre-trained on pure text corpus and thus shows difficulty in understanding the pathology visually.

## A.4  MORE VISUALIZATIONS

In Fig. 8, we demonstrate how the image registration takes effect. Without image registration (first row), the gold reference image is usually not well-aligned with the prior image according to the

Table 5: Additional ablation studies on the effect of different text encoders including BiodmedCLIP, CLIP and PubMedBERT.

| Inst. | Text Encoder | Pathology AUC | Race AUC | Age Pearson Corr. | Avg |
|-------|-------------|---------------|----------|-------------------|-----|
| Imp. | PubMedBERT | 74.93 | 98.75 | 83.48 | 81.17 |
| GPT4 | PubMedBERT | 74.54 | 98.90 | 83.9 | 81.08 |
| Imp. | CLIP | 76.37 | 98.82 | 83.28 | 81.88 |
| GPT4 | CLIP | 76.67 | 98.75 | 83.05 | 82.02 |
| Imp. | BiomedCLIP | 78.06 | 98.67 | 82.73 | 82.70 |
| GPT4 | BiomedCLIP | 78.22 | 98.70 | 82.96 | 82.83 |

Figure 8: Image registration comparison. Top row, from left to right: Image 1 (original, before registration), Image 2 (registered on Image 1), and the difference (delta) between the two. Bottom row mirrors the top but highlights the registered images and their differences. The delta images emphasize the areas of change and alignment.

heatmap in the last column, because they are taken independently at two different time periods. To calibrate the two images, we apply a registration algorithm to the reference image using the prior image as the anchor. Afterward, the misalignment is significantly mitigated as shown on the bottom-right image.

In Fig. 9, we show the medical journey of a patient given the prior image as shown in the first column. We use different text prompts to impose the pathology at different magnitudes, such as 'slight', 'moderate' and 'large'. Surprisingly, our *MedJourney* strictly follows the instructions and makes appropriate changes to the prior image, which exhibits a smooth and natural pathology progression.

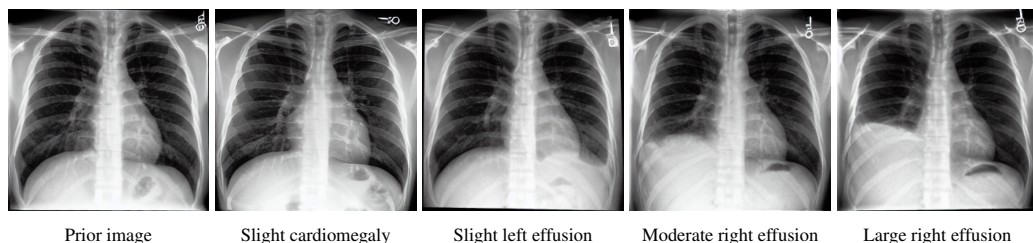

| Prior image | Slight cardiomegaly | Slight left effusion | Moderate right effusion | Large right effusion |

Figure 9: Patient medical journey emulated by our *MedJourney*. Given a prior image, *MedJourney* can generate target images that precisely reflect the progressions described in the text prompts.

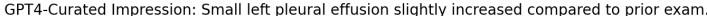

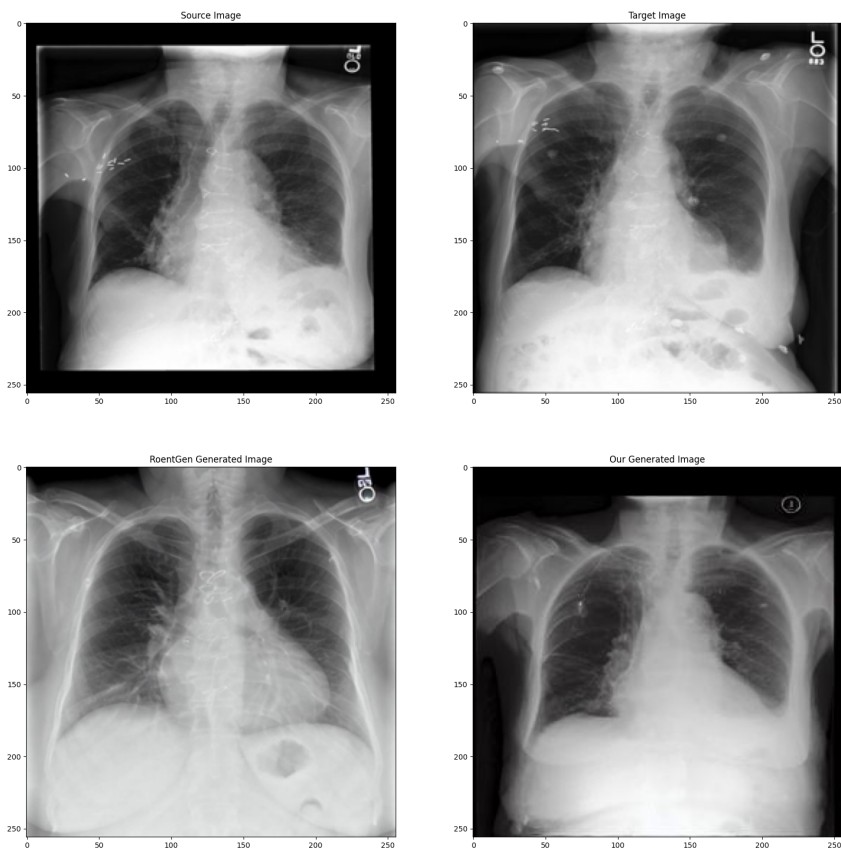

Figure 10: Top left: prior image. Top right: reference image. Bottom left: RoentGen. Bottom right: MedJourney (Ours). *Prompt: Small left pleural effusion slightly increased compared to prior exam.*

PT4-Curated Impression: In comparison with the previous study, low lung volumes are again noted. There is an increasing right pleural effusion
ith increased prominence of the cardiac silhouette. No evidence of left pleural effusion. Increased prominence of centra
l pulmonary vessels is observed, with no evidence of peripheral venous congestion.

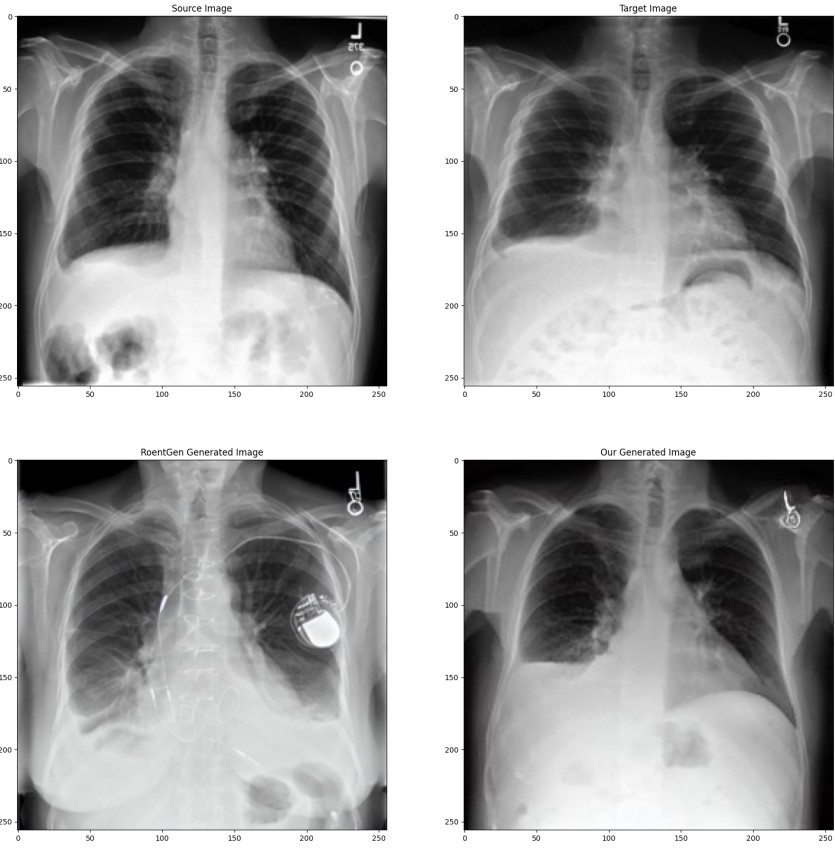

Figure 11: Top left: prior image. Top right: reference image. Bottom left: RoentGen. Bottom right: MedJourney (Ours). *Prompt: In comparison with the previous study, low lung volumes are again noted. There is an increasing right pleural effusion with increased prominence of the cardiac silhouette. No evidence of left pleural effusion. Increased prominence of central pulmonary vessels is observed, with no evidence of peripheral venous congestion.*

GPT4-Curated Impression: Moderate size right pleural effusion with right basilar opacity, likely representing compressive atelectasis. Infection cannot be excluded.

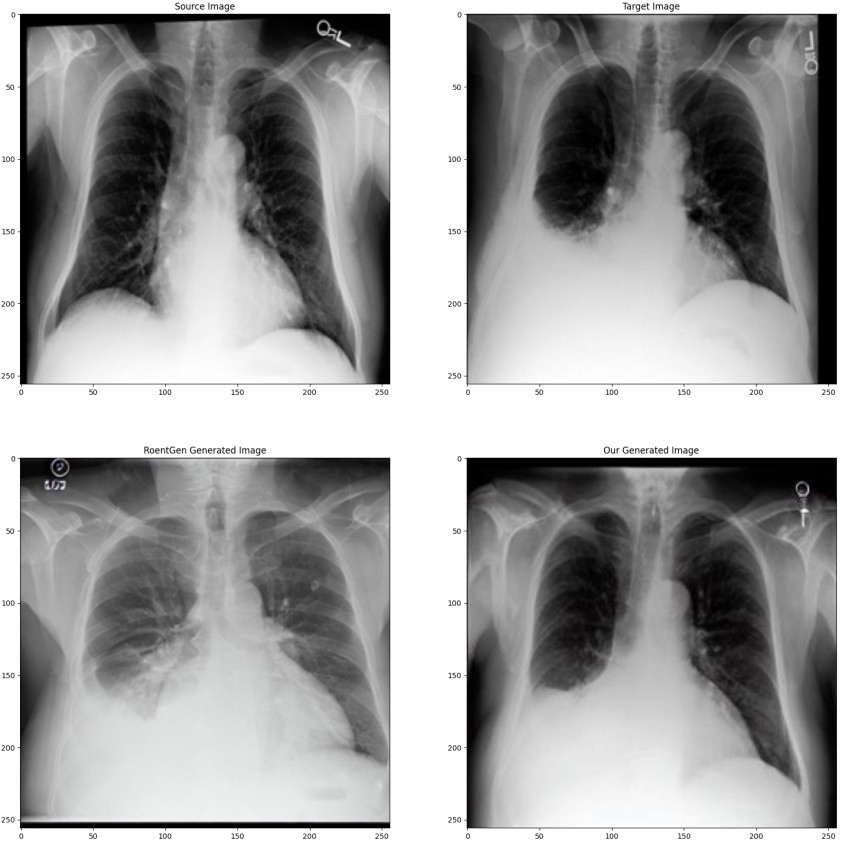

Figure 12: Top left: prior image. Top right: reference image. Bottom left: RoentGen. Bottom right: MedJourney (Ours). *Prompt: Moderate size right pleural effusion with right basilar opacity, likely representing compressive atelectasis. Infection cannot be excluded.*

4-Curated Impression: 1. Stable moderate to large right apical pneumothorax with unchanged extensive subcutaneous gas. Chest tube holes ar
ntained within pneumothorax, but tip terminates within the soft tissues of the thoracic inlet.
2. Right lower lobe opaci
fication, likely due to aspiration.

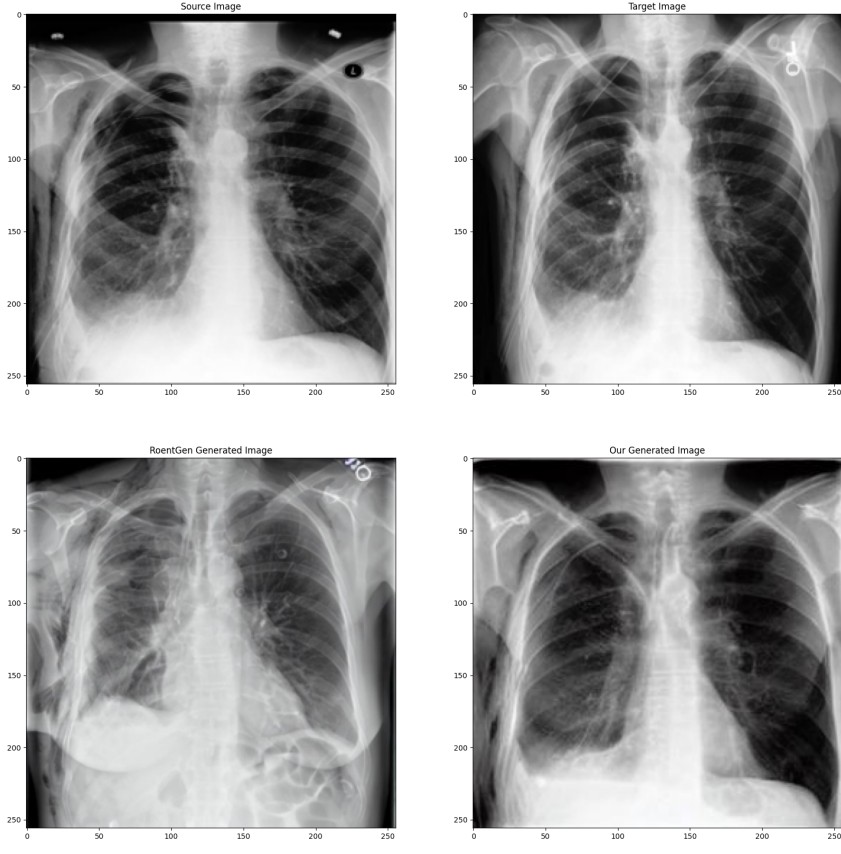

Figure 13: Top left: prior image. Top right: reference image. Bottom left: RoentGen. Bottom right: MedJourney (Ours). *Prompt: 1. Stable moderate to large right apical pneumothorax with unchanged extensive subcutaneous gas. Chest tube holes maintained within pneumothorax, but tip terminates within the soft tissues of the thoracic inlet. 2. Right lower lobe opacification, likely due to aspiration.*

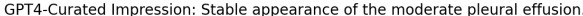

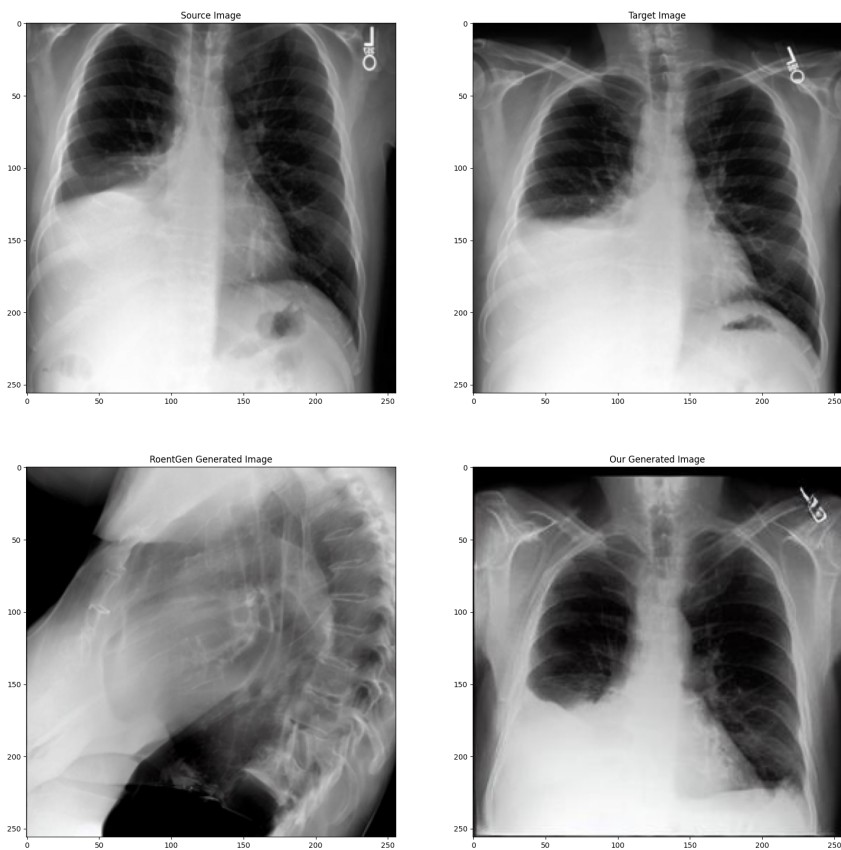

Figure 14: Top left: prior image. Top right: reference image. Bottom left: RoentGen. Bottom right: MedJourney (Ours). *Prompt: Stable appearance of the moderate pleural effusion.*

GPT4-Curated Impression: No significant interval change. Unchanged bilateral pleural effusions, right greater than left.

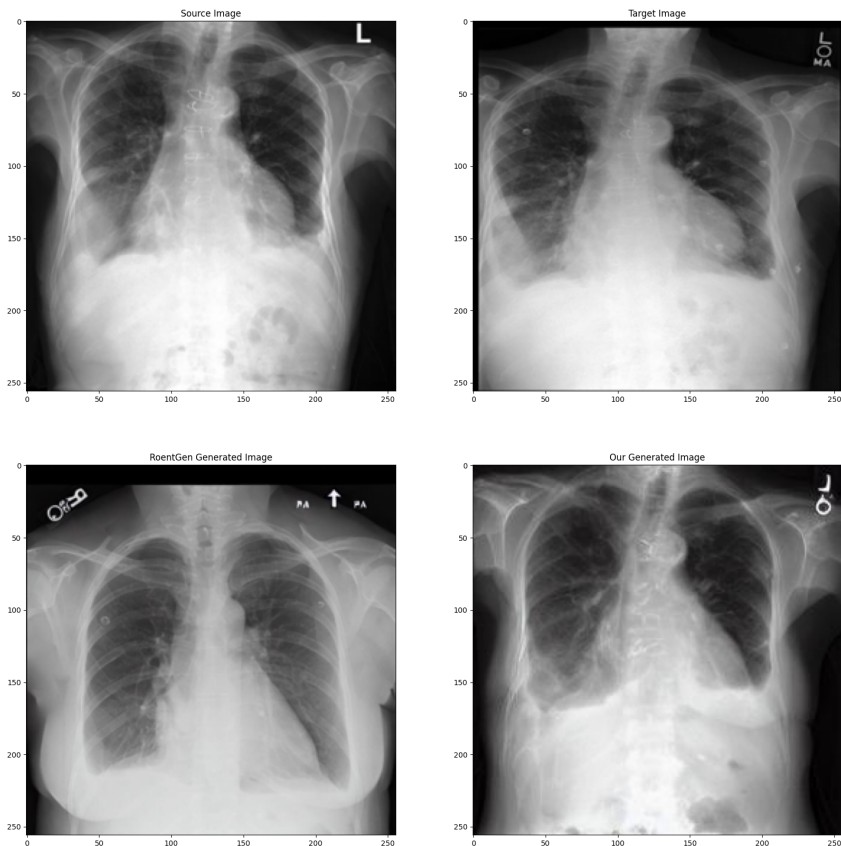

Figure 15: Top left: prior image. Top right: reference image. Bottom left: RoentGen. Bottom right: MedJourney (Ours). *Prompt: No significant interval change. Unchanged bilateral pleural effusions, right greater than left.*

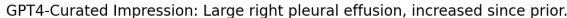

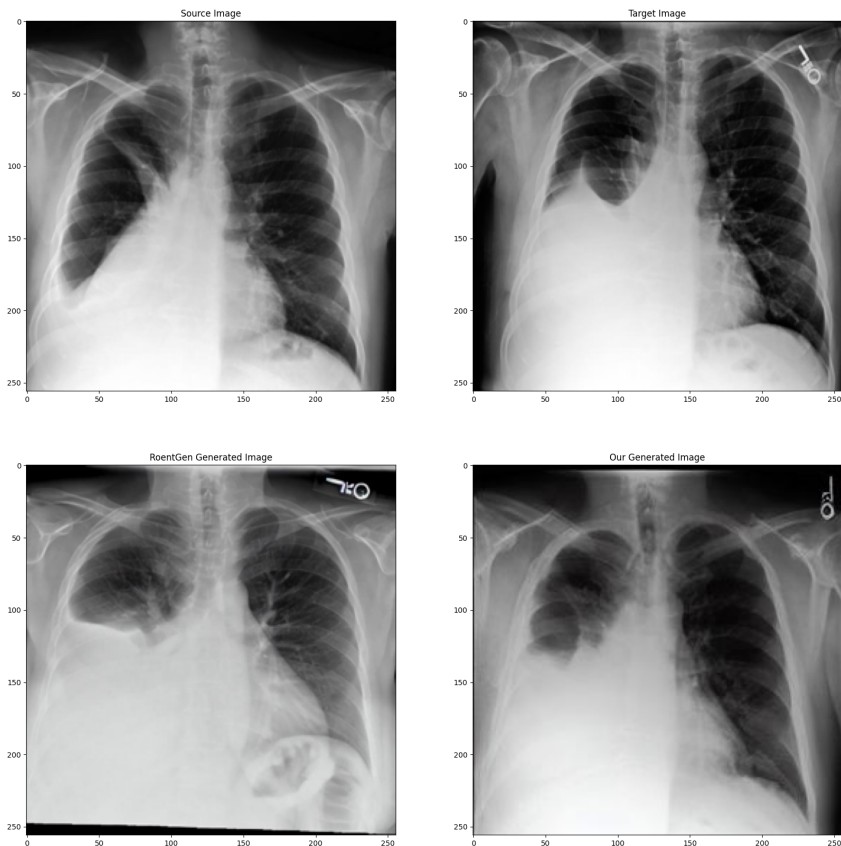

Figure 16: Top left: prior image. Top right: reference image. Bottom left: RoentGen. Bottom right: MedJourney (Ours). *Prompt: Large right pleural effusion, increased since prior.*

