# OpenReview forum: "MedJourney: Counterfactual Medical Image Generation by Instruction-Learning from Multimodal Patient Journeys"
_ICLR.cc/2024/Conference — Submitted to ICLR 2024_

### Official Review · Reviewer_egz5 · 2023-10-31

**Soundness:** 3 good
**Presentation:** 3 good
**Contribution:** 3 good
**Rating:** 6
**Confidence:** 3

**Summary:**

This article proposes a counterfactual medical image generation method that can generate high-quality counterfactual images guided by natural language. To achieve this, the authors use multimodal medical imaging information such as sequential images and medical reports to generate progress descriptions of image sequences using GPT-4. Based on the sequential images, reports, and these progress descriptions, a two-stage training diffusion model is used for image generation.

**Strengths:**

1. A method is proposed to generate counterfactual medical images based on natural language descriptions.
2. Curriculum learning is introduced to make full use of the ability of non-sequential images to enhance the base model.
3. GPT-4 is used to process textual information about disease progression in the training set and generate intermediate progress descriptions. Detailed and feasible descriptions are provided for constructing the training set.
4. The ability of generating counterfactual medical images is evaluated on multiple test sets including pathology, race, age, spatial alignment, etc.

**Weaknesses:**

1. The corresponding reference image in the training set should be included in Figure 4.
2. The accuracy of descriptions generated by GPT-4 lacks evaluation, which directly affects whether the generated images are correct or not.

**Questions:**

1. Why use the word `counterfactual` instead of direct `image editing`? The latter is more intuitive and understandable. Since reference images already exist in this case, such image generation does not appear counterfactual.
2. In Table 4, MedJourney has more accurate segmentation results than Reference Image; does this mean that segmentation results are difficult to evaluate the quality of generated images?

---

> ### Author Response · Authors · 2023-11-23
>
> Thanks for your insightful comments and suggestions. Below we will address all the questions and concerns.
>
> # Re adding reference image in Figure 4:
>
> Thanks for the suggestion. We have the reference image in Figure 5, second from left on top row. We will add it to Figure 4 in the final version.
>
> # Re GPT-4 evaluation:
>
> Although GPT-4 has no specialized training in biomedicine, many prior studies have demonstrated its impressive capabilities in understanding and processing biomedical text. E.g., book by Peter Lee et al. on: “The AI Revolution in Medicine: GPT-4 and Beyond.” offers insights into this aspect.
>
> Additionally, one of the coauthors, a board-certified practicing radiologist at a top US medical school, has played a crucial role in assessing the utility of GPT-4 in our context. The progression description generated by GPT-4 appears reasonable upon preliminary examination. We will add more systematic assessments in the final version.
>
> To illustrate, here are some examples reviewed by the radiologist, comparing GPT-4 vs orginal text:
> |Impression| GPT-4 |
> |-----| -----|
> “Resolving right middle lobe pneumonia. A followup chest radiograph in 4 weeks is recommended. If the right middle lobe opacity fails to completely resolve by that time, a chest CT should be performed at that time to exclude an endobronchial lesion. New small right pleural effusion.” |”Resolving right middle lobe pneumonia. New small right pleural effusion.”|
> “Subtle right lower lobe opacity may represent early pneumonia. These findings were discussed with Dr. _ by Dr. _ at 2:30 p.m.” |"Subtle right lower lobe opacity may represent early pneumonia."
>
> These examples demonstrate GPT-4's ability to succinctly summarize key radiological findings, which aligns well with clinical needs and workflow efficiency. This aspect will be further elaborated in the final version of our manuscript.
>
> Likewise, in addition to the quantitative evaluation, we have conducted several review sessions with the aforementioned radiologist co-author, who expressed satisfaction with the image generation quality. We will mention this in the final version. Thanks much for your suggestion.
>
>
> # Re counterfactual vs image editing:
>
> We agree that the underlying technique might share a lot of similarity, but the real-world semantics are quite different. In biomedicine, a clinician won’t be thinking about this generation problem as editing an image. Instead, their interest lies in counterfactual reasoning,  i.e., projecting what if the patient’s condition has progressed in certain ways. Many prior works have studied the “Counterfactual” factor for medical images, e.g. Cohen et, al. “Gifsplanation via Latent Shift: A Simple Autoencoder Approach to Counterfactual Generation for Chest X-rays”; Sumedha Singla, et, al. “Explaining the Black-box Smoothly- A Counterfactual Approach”. In this context, counterfactual just means answering what if questions, which could be with or without the prior image. However, in the case of progression modeling, the patient already has some prior image taken. So it is natural to condition on the prior image, rather than generating a newmage from scratch without any constraints. This distinction is pivotal in our research, as it aligns with the clinical workflow and decision-making processes, emphasizing the practical applicability in real-world medical settings. We will mention this in the final version.
>
>
> # Re Table 4:
>
> Thanks for the insightful comment. Here, we are comparing the similarity of a generated image with the prior image. The reference image may have lower DICE score due to variations in the creation process for the consecutive images, which are taken at different times, potentially by different clinicians with different machines. This variation is a typical characteristic in clinical imaging and does not imply inferiority of the reference image.
>
> By contrast, the extremely low scores for baseline systems indicate that they are completely off base with respect to the prior image.
>
> We will clarify this in the final version.

---

### Official Review · Reviewer_jLph · 2023-11-01

**Soundness:** 3 good
**Presentation:** 3 good
**Contribution:** 3 good
**Rating:** 6
**Confidence:** 4

**Summary:**

The paper introduces a method for counterfactual medical image generation leveraging instruction-learning from multimodal patient journeys. Amidst rapid advancements in image editing using natural-language instruction, as seen with "InstructPix2Pix," there remains a significant gap in biomedicine. Current models, though effective in generic contexts, fall short in the biomedical domain. MedJourney addresses this by generating counterfactual medical images based on instruction-learning from patient data. The method involves processing two medical images taken at different time points, using GPT-4 to generate a natural language description of disease progression based on associated reports. This information is used to train a latent diffusion model for image generation. Due to the limited availability of image time-series data, the authors employ a two-stage curriculum: first pretraining with abundant single image-report pairs, followed by training with counterfactual triples. Experiments on the MIMIC-CXR dataset reveal MedJourney's superior performance over existing methods like InstructPix2Pix and RoentGen.

**Strengths:**

Strengths:
1. **First paper to develop models for counterfactual image generation in the medical domain**:
The paper emphasizes the largely unexplored domain of unconstrained counterfactual medical image generation. While there have been attempts in the biomedical imaging field to answer "what if" questions, existing methods primarily focus on simple image class changes, akin to predefined class edits. MedJourney stands out as it closely follows arbitrary natural-language descriptions of disease progression to generate counterfactual images, making it a pioneering method in the field.

2. **The innovative use of GPT-4 to extract the progression description**:
MedJourney uniquely leverages the GPT-4 model to generate a natural language description of disease progression. When given two medical images from different time points of a patient, manually synthesizing image-editing instructions from their reports can be both expensive and time-consuming. Instead of manual synthesis, the authors utilize GPT-4 to generate these descriptions automatically from the imaging reports, offering a scalable solution.

3. **Benchmarked against general domain method: InstructPix2Pix**:
In the rapidly evolving field of image editing with natural language instruction, InstructPix2Pix has been a benchmark. However, models like InstructPix2Pix, trained on generic images and text, might not be optimally suited for the nuanced biomedical domain. MedJourney, in its extensive tests on the MIMIC-CXR dataset, not only compared its performance with methods tailored for the medical domain like RoentGen but also with general domain methods like InstructPix2Pix. The results showed that MedJourney substantially outperforms these state-of-the-art methods in both instruction image editing and medical image generation.

4. **Authors gave an honest attempt at establishing quantitative evaluation metrics**:
Recognizing the complexity and challenges associated with evaluating counterfactual medical image generation, the authors introduced an extensive suite of tests. These tests include evaluations based on various factors such as pathology, race, age, and spatial alignment. This can be certainly improved upon but it is at least a step in the right direction.

**Weaknesses:**

Areas for improvement:

1. **Comparison to more recent works that closely follow the instruction and reference image**:
While MedJourney has shown promise in counterfactual medical image generation, it lacks a direct comparison with more recent works, such as "Imagic: Text-Based Real Image Editing with Diffusion Models." Imagic, designed to closely follow instructions and reference images, is potentially better suited for the tasks that MedJourney addresses with medical images. A side-by-side comparison or integration of insights from Imagic could potentially strengthen the robustness and applicability of MedJourney.

2. **Need for reader study in addition to quantitative metrics for qualitative evaluation of different methods**:
While the authors have introduced a comprehensive suite of quantitative tests, the paper could benefit from a reader study that provides qualitative insights. Direct feedback from radiologists or medical experts on the interpretability, authenticity, and clinical utility of the counterfactual images would enhance the evaluation depth and offer a more holistic understanding of MedJourney's real-world applicability.

3. **Ethical considerations and limitations are not discussed**:
The application of AI in the medical domain has inherent ethical implications. Given that counterfactual medical images could influence clinical decisions, the absence of a discussion on the ethical considerations and potential risks associated with MedJourney is a significant oversight. Addressing these concerns would not only ensure the safe deployment of such methods but also enhance trust among medical practitioners and patients.

4. **How well does it apply to other medical image datasets?**
While MedJourney's performance on the MIMIC-CXR dataset is commendable, its generalizability across diverse medical imaging modalities remains uncertain. Demonstrating its efficacy on a broader range of datasets, such as MRI, CT scans, or ultrasounds, would substantiate its versatility. Furthermore, insights into how the method adapts and performs across different diseases, anatomical regions, or imaging techniques would solidify its position as a universally applicable counterfactual medical image generation tool.

5. **Minor edits**: All citations should be in parenthesis. In many place, the parenthesis are missing which messes up the main text. Please correct those.

6. **Demographic diversity is crucial for medical datasets**:
Medical datasets must accurately represent the diverse patient populations they serve to ensure that AI models generalize well across varied demographic groups. The authors acknowledge in the appendix that the dataset has an overrepresentation of the white elderly population. Such biases can inadvertently lead to models that perform sub-optimally or even erroneously for underrepresented groups, which can have significant clinical implications. Highlighting this limitation in the main body of the paper, especially in sections like 'Limitations' or 'Discussion,' would underscore the importance of demographic diversity. It would also serve as a call to action for researchers to actively pursue more balanced datasets, ensuring equitable healthcare outcomes for all patients regardless of their background.

7. **Add FID as an additional metric**: Please add FID as an additional qualitative metric to evaluate different methods. See UnixGen and RoentGen paper for details on the FID metric.

**Questions:**

If authors address the areas of improvement, I would be open to update my score.

**Details Of Ethics Concerns:**

1. **Demographic diversity is crucial for medical datasets**:
Medical datasets must accurately represent the diverse patient populations they serve to ensure that AI models generalize well across varied demographic groups. The authors acknowledge in the appendix that the dataset has an overrepresentation of the white elderly population. Such biases can inadvertently lead to models that perform sub-optimally or even erroneously for underrepresented groups, which can have significant clinical implications. Highlighting this limitation in the main body of the paper, especially in sections like 'Limitations' or 'Discussion,' would underscore the importance of demographic diversity. It would also serve as a call to action for researchers to actively pursue more balanced datasets, ensuring equitable healthcare outcomes for all patients regardless of their background.

2. **Ethical considerations and limitations are not discussed**:
The application of AI in the medical domain has inherent ethical implications. Given that counterfactual medical images could influence clinical decisions, the absence of a discussion on the ethical considerations and potential risks associated with MedJourney is a significant oversight. Addressing these concerns would not only ensure the safe deployment of such methods but also enhance trust among medical practitioners and patients.

---

> ### Author Response · Authors · 2023-11-23
>
> Thanks for your insightful comments and suggestions. Below we will address all the questions and concerns.
>
> # Re comparison to Imagic:
>
> Thanks for this suggestion. Imagic requires per-example inversion and fine-tuning. Namely, for each input image and target description, it requires a fine-tuning process to obtain an optimal text embedding close to the target description, fine-tune the diffusion model conditioned on the optimal text embedding (rather than noise in the original diffusion model), and then apply the newly fine-tuned diffusion model to an interpolation of original & target text embeddings to generate the new image. This is very cumbersome and not very generalizable.
>
> By contrast, our method builds on the more recent InstructPix2Pix method, which learns a single model that can generate new image in a single forward pass, without requiring elaborate text embedding and diffusion learning for each test example as in Imagic.
>
> We will add discussion to explain why our method is superior to the Imagic approach.
>
>
> # Re reader study by radiologist:
>
> One of the coauthors is a board-certified practicing radiology at a top US medical school. In addition to the quantitative evaluation, we have conducted several review sessions with this radiologist co-author, who expressed satisfaction with the generation quality. We will mention this in the final version. Thanks much for your suggestion.
>
>
> # Re ethical considerations and limitations:
>
> Thanks much for this insightful suggestion. We have discussed limitations in the Appendix. But as you suggested, this is such an important aspect that we will move our existing discussion on limitations to the main body and add additional considerations for ethical implications.
>
>
> # Re generalization to other medical image datasets:
>
> Thanks much for this suggestion. While we tried to break new ground in counterfactual medical image generation, this is just the first baby step towards a universally applicable tool. We plan to explore exactly along the directions you have outlined in future work.
>
> # Re citation:
>
> Thanks for the good catch! We will fix them.
>
> # Re demographic diversity:
>
> Thanks much for this insightful suggestions. We will move our discussion about demographic diversity to the main body.
> Further, we have conducted additional evaluations focusing specifically on racial diversity. Our analysis compared image generations for the dominant demographic group (white) and an underrepresented group (black). Despite the training data itself being skewed in terms of racial representation (**17%** black vs **78%** white), our results are promising. MedJourney successfully retained the racial features of the prior image with high fidelity, scoring **0.92** for the underrepresented group and **0.98** for the dominant group (we used the SOTA image race classifier, as mentioned in sec 4.3, as evaluator and AUC as metric). This indicates our MedJourney has a decent property of preserving demographic diversity,  and further suggests that it may contribute to mitigating racial representation disparities through data augmentation for underrepresented groups.
> While our results are promising, they also emphasize the crucial need for balanced datasets in medical research. We will include this insight in the final version.
>
> # Re adding FID metric:
>
> Thanks for the suggestion. We have incorporated this metric into our analysis and will present it in the final version. The FID results are as follows:
> | Method| FID|
> |-|-|
> | Stable Diffusion| 291.13
> |InstructPix2Pix| 180.72
> |RoentGen| 42.61
> |MedJourney| **29.68**
>
> We believe these results further demonstrate the effectiveness of MedJouney.

---

### Official Review · Reviewer_A6Mk · 2023-11-01

**Soundness:** 3 good
**Presentation:** 3 good
**Contribution:** 2 fair
**Rating:** 5
**Confidence:** 4

**Summary:**

The paper addresses the challenge of generating counterfactual medical images. It involves a two-step process: initially, it utilizes GPT-4 to create a natural language narrative describing the progression of a disease based on two medical reports corresponding to images taken at different times. Subsequently, the paper employs a latent diffusion model to produce the counterfactual medical images. The experimental evaluation using MIMIC-CXR demonstrates a significant improvement over previous methods, particularly in aligning pathology, race, and age in the generated counterfactual images.

**Strengths:**

- The paper investigates the unique problem of counterfactual image generation, generating counterfactual images from description of disease progression. It introduces a curriculum learning technique for generating counterfactual images using these progression descriptions. This approach is novel and would interest the machine learning and medical imaging community.

- The paper is well-structured, making it straightforward and easy to follow.

- The methodology used for evaluating pathology, race, and age classification, along with the CMIG score that measures both accuracy and feature retention, provides a practical and insightful perspective on the issue.

**Weaknesses:**

The paper's enhancement primarily hinges on text processing and image registration, both of which raise concerns:

- **Concerns with Text Processing using GPT-4**:
    - **Validity of Disease Progression**: he paper does not indicate whether the disease progression generated by GPT-4 has been compared or verified against radiologists' opinions. Such a comparison is crucial for establishing the reliability of the generated progression description.
    - **Lack of Sanity Checks**: There seems to be no mention of sanity checks for the progression generated between two images. Considering GPT-4's capability to generate progression even for unrelated images of different patients, it's vital to understand how the paper ensures the accuracy and relevance of these progressions.
    - **Comparison between Impression Section and GPT-4 Usage**: The advantages of using GPT-4 over the impression section in reports are minimal (as evident in Table 2). Clarification on how the impression section was used and whether GPT-4 is actually essential for this study would be beneficial, given the marginal gains across metrics.

- **Image Registration Contribution**:
    - The substantial improvements seem to stem from image registration, primarily leveraging the SimpleITK toolkit (Beare et al. 2018). This raises concerns about the originality of the MedJourney method's contributions. An expanded discussion on how image registration was adapted for the proposed method would provide a clearer understanding of its impact.

Additional Question:

- Concerning the pathology classifier, it's unclear which report's reference pathology labels are used. Are these labels compared with those generated in Cohen et al. 2021, which are used for assessing the counterfactual image model? Clarification on this process would be helpful.

- **Prompt Requirement**: The necessity of the prompt “a photo of chest x-ray” is not justified. Including results without this prompt might offer more insight into its impact.

Minor Issues:
- There's a missing citation for GPT-4 in the related work section on page 3.
- Figure 3 could be better placed in the experiments section, as it is not referenced beforehand.
- Figure 1 is not referenced in the text.
- Paper uses Figure and Fig. interchangeably.

**Questions:**

Kindly refer to the concerns raised in the above section.

---

> ### Author Response · Authors · 2023-11-23
>
> Thanks for your insight comments and suggestions. Below we will address all the questions and concerns.
>
> # Re concerns with text processing:
>
> **Validity of disease progression**: Thanks for the suggestion. One of the coauthors is a board-certified practicing radiology at a top US medical school. The progression description generated by GPT-4 appears reasonable upon preliminary examination. We will add more systematic assessments in the final version. In general, although GPT-4 has no specialized training in biomedicine, many prior studies have demonstrated its amazing capabilities in understanding and processing biomedical text. E.g., see the book by Peter Lee et al. on: “The AI Revolution in Medicine: GPT-4 and Beyond.”
>
> **Lack of sanity check**: In addition to the quantitative evaluation, we have conducted several review sessions with the aforementioned radiologist co-author, who expressed satisfaction with the generation quality. We will mention this in the final version. Thanks much for your suggestion.
>
> **Comparison between Impression and GPT-4**: The impression section is taken from the gold report describing the target image. The small difference between using impression vs GPT-4 generated progression may stem from the characteristics of this dataset, where in many cases the first image is normal, and thus the pathological description of the follow-up image essentially describes progression. Additionally, some impression would include comparison against the prior image, also similar to what GPT-4 would generate as progression. As in 5.3 “impression v.s. GPT-4 generation”, GPT-4 generated is usually more succinct (averaging 146 characters v.s. 179) and more naturally phrased. For example,
> |Impression| GPT-4 |
> |-----| -----|
> “Resolving right middle lobe pneumonia. A followup chest radiograph in 4 weeks is recommended. If the right middle lobe opacity fails to completely resolve by that time, a chest CT should be performed at that time to exclude an endobronchial lesion. New small right pleural effusion.” |”Resolving right middle lobe pneumonia. New small right pleural effusion.”|
> “Subtle right lower lobe opacity may represent early pneumonia. These findings were discussed with Dr. _ by Dr. _ at 2:30 p.m.” |"Subtle right lower lobe opacity may represent early pneumonia."
>
> # Re image registration contribution:
>
> Note that image registration is only a preprocessing step that helps align the two consecutive images for a given patient in the training data. This is a standard technique to reduce immaterial variations of images taken at different times by different technicians, e.g., translational shift or rotation. Image registration alone can’t perform counterfactual image generation and is thus irrelevant with respect to the novelty of MedJourney. We will clarify this in the final version.
>
> A significant challenge in our research was the limited dataset size (approximately 10,000 triplets), posing the risk of learning spurious correlations due to minor, non-essential variations in images. By implementing image registration, we effectively minimized these immaterial variations, enabling the model to focus on material changes indicative of disease progression.
>
> This image registration procedure is used merely for preprocessing training data. This procedure is illustrated in Appendix Figure 8, and to enhance understanding, we will move it to the main part in the final version.
>
> This innovative use of image registration in our specific learning pipeline is, to our knowledge, unprecedented in both generic and medical image generation research. We want to reiterate that, image registration cannot generate counterfactual image on its own, its use in MedJourney represents an inventive adaptation of existing techniques, augmenting rather than diminishing the method's originality.

---

> > ### Author Response · Authors · 2023-11-23
> >
> > To continue.
> >
> > # Re pathology classifier:
> >
> > In counterfactual image generation, the goal is to generate a new image given the original image and the description of disease progression. Therefore, the evaluation tries to assess whether the new image indeed matches the gold image in pathological findings and other key semantics. The report associated with the gold image is used to derive reference pathology labels.
> >
> > E.g., given consecutive Image 1 / Report 1 and Image 2 / Report 2, MedJourney will input Image 1 and the delta report, and generate Image 2’. We’ll then evaluate Image 2’ based on Report 2.
> >
> > Specifically, we ran the image-based classifier from Cohen et al. 2020b on Image 2’ to obtain pathological findings based on the generated image, and then compare them against pathological findings obtained from Report 2 using text-based CheXpert (which was used to train the image-based classifier in Cohen et al. 2020b).
> >
> > We did catch a typo in our reference: “Cohen et al. 2021” should be “Cohen et al. 2020b”. We will fix this in the final version.
> >
> > Joseph Paul Cohen, Mohammad Hashir, Rupert Brooks, and Hadrien Bertrand. On the limits of
> > cross-domain generalization in automated X-ray prediction. Medical Imaging with Deep Learning, 2020b. URL [https://arxiv.org/abs/2002.02497](https://arxiv.org/abs/2002.02497).
> >
> > # Re prompt requirement:
> >
> > Thanks for pointing out. This prompt “a photo of chest x-ray” facilitates to generate better results in our exploration. As we observed in our experiments, without the prompt, the baseline Stable Diffusion (SD) model tends to produce arbitrary results, getting lost from the descriptions of disease progression. That being said, we introduced this prompt to elicit the maximal capacity of SD for conducting the fair comparison as possible as we can. We will include quantitative results in the final version when this prompt is not used for the justification.
> >
> > # Re issue on citations and figures:
> >
> > Thank you very much for all the good catches. We will fix them in the final version.

---

### Author Response · Authors · 2023-11-23
**To Area Chairs and all Reviewers**

First of all, we thank all reviewers for their valuable comments and suggestions!
We sincerely appreciate all reviewers’ time and efforts in reviewing our paper. We are glad to find that reviewers generally recognized our contributions:
## Novelty and Methodology.
-	First to develop models for unconstrained counterfactual medical image generation in the medical domain, marking a pioneering approach in the field (jLph31).
-	Introduces a novel curriculum learning technique for counterfactual image generation based on disease progression descriptions, appealing to the machine learning and medical imaging community (A6Mk01).
-	Innovative use of GPT-4 for efficient and scalable generation of natural language descriptions of disease progression (jLph31).
-	Proposed method incorporates curriculum learning to enhance model performance using non-paired images (egz5).
## Evaluation and Performance.
-	Comprehensive evaluation methodology for pathology, race, and age classification, along with the introduction of the CMIG score (A6Mk01).
-	Demonstrates superior performance in instruction image editing and medical image generation compared to state-of-the-art methods like InstructPix2Pix and RoentGen (jLph31).
-	Extensive suite of tests including evaluations based on pathology, race, age, and spatial alignment, indicating a thorough approach (jLph31).
## Structure and Clarity.
-	Paper is well-structured, making it straightforward and easy to follow (A6Mk01).
-	Detailed and feasible descriptions provided for constructing the training set, enhancing clarity and feasibility (egz5).

## New Qualitative and Quantitative Results.

1. **Enhanced Illustrative Examples with GPT-4's Role:**
   We have added detailed qualitative examples to better demonstrate GPT-4 vs Original Impression of image report: GPT-4 generated tend to be more succinct (averaging 146 characters v.s. 179) and more naturally phrased. For example,

   | Impression | GPT-4 |
   | ------- | -------------------- |
   | “Resolving right middle lobe pneumonia. A follow-up chest radiograph in 4 weeks is recommended. If the right middle lobe opacity fails to completely resolve by that time, a chest CT should be performed at that time to exclude an endobronchial lesion. New small right pleural effusion.” | “Resolving right middle lobe pneumonia. New small right pleural effusion.” |
   | “Subtle right lower lobe opacity may represent early pneumonia. These findings were discussed with Dr. _ by Dr. _ at 2:30 p.m.” | "Subtle right lower lobe opacity may represent early pneumonia." |


2. **Introduction of FID Metric for Comparative Analysis:**
   We have incorporated the Frechet Inception Distance (FID) metric to provide a quantitative comparison with baseline models. This metric demonstrates the superiority of our model, MedJourney, in generating high-fidelity medical images.

   | Method          | FID Score |
   | --------------- | --------- |
   | Stable Diffusion| 291.13    |
   | InstructPix2Pix | 180.72    |
   | RoentGen        | 42.61     |
   | MedJourney      | **29.68** |

   The significantly lower FID score for MedJourney indicates its advanced capability in producing more accurate and realistic medical imagery.

3. **Evaluation Focusing on Demographic Diversity:**
   Inspired by Reviewer jLph31‘s concern on demographic disparity for medical datasets, we have evaluated our model's performance among overrepresented and underrepresented ethnic groups. The results indicate MedJourney's decent property of preserving demographic diversity, and emphasize the crucial need for balanced datasets in medical research.

   | Ethnicity           | Training Data Percentage | Race Retention AUC |
   | ------------------- | ------------------------ | ------------------ |
   | Black               | 17%                      | 0.92               |
   | White               | 78%                      | 0.98               |


## Reader Study/Sanity Check by Radiologist:
Both reviewers A6Mk and jLph raised concerns regarding the absence of a reader study or sanity check by a radiologist for the GPT-4 disease progression generation and MedJourney image generation. To address the concerns, we highlight the involvement of a board-certified radiologist, who is also a coauthor and affiliated with a top US medical school. In addition to the quantitative evaluation, we have conducted several review sessions, assessing GPT-4's progression descriptions and MedJourney's image generation quality. The radiologist has confirmed the reasonableness of GPT-4's outputs and expressed satisfaction with the quality of images generated by MedJourney upon preliminary examination.

We carefully read all comments and attempted to address the concerns with comprehensive responses, and hope our responses could answer the questions and resolve any concerns regarding our work. Again, we thank all reviewers again for their efforts and constructive comments!

---

### Meta-Review · Area_Chair_75gC · 2023-12-06

**Metareview:**

Authors propose a prompt-based counterfactual image generation for medical data, where they use GPT model to prepare before/after prompts to learn the mappings.

**Justification For Why Not Higher Score:**

Reviewers agree that the results have been checked subjectively by a expert coauthor, which is not very reliable given the sensitivity of the data. Furthermore Imagic has been discussed as an existing approach that authors did not compare against to show the superiority of their proposed technique.

**Justification For Why Not Lower Score:**

N/A

---

### Decision · Program_Chairs · 2024-01-16

Reject